# The relationship between early life urbanicity and depression in late adulthood: evidence from the Survey of Health, Ageing and Retirement in Europe

Daniel Howdon,[1,2] Jochen Mierau,[2,3] Samuel Liew[2,4]

¹Academic Unit of Health Economics, University of Leeds, Leeds Institute of Health Sciences, Leeds, UK
²Economics, Econometrics and Finance, Rijksuniversiteit Groningen Faculteit Economie en Bedrijfskunde, Groningen, The Netherlands
³Aletta Jacobs School of Public Health, Rijksuniversiteit Groningen, Groningen, The Netherlands
⁴Department of Economics, Copenhagen Business School, Frederiksberg, Denmark

**Correspondence to**
Dr Jochen Mierau;
j.o.mierau@rug.nl

## ABSTRACT

**Objectives** We aimed to study the association of childhood urbanicity with depressive symptoms in late adulthood.

**Design, setting and participants** We used linear and logistic regressions to analyse data drawn from 20 400 respondents from the Survey of Health, Ageing and Retirement in Europe, a panel dataset incorporating a representative sample of the 50+ population in 13 European countries.

**Outcomes and analysis** Childhood urbanicity was determined using self-reports of the respondents' circumstances at age 10, and late-adulthood depression using the EURO-D scale. We conditioned on circumstances early in life as well as later in life, most importantly late-adulthood urbanicity. We estimated the associations using linear regression models and limited dependent variable models.

**Results** A pooled regression of both men and women suggested that childhood urbanicity is associated non-monotonically with depression in late adulthood and is particularly apparent for those spending their childhoods in suburban settings. We found that individuals who spend the longest time in their childhood in a suburban home exhibit an average increase in probability of 3.4 (CI 1.1 to 5.7) percentage points in reporting four or more depressive symptoms. The association was robust to the inclusion of a host of household characteristics associated with childhood urbanicity and was independent of current urbanicity and current income. When broken down by gender, we found some evidence of associations between depressive outcomes and urban living for men, and stronger evidence of such associations with urban and suburban living for women who exhibit an increase of 5.6 (CI 2.2 to 9.0) percentage points in reporting four or more depressive symptoms.

**Conclusions** Our analysis reveals a relationship between childhood urbanicity and depression in late adulthood. The evidence presented on the nature of this relationship is not straightforward but is broadly suggestive of a link, differing by gender, between greater urbanicity and higher levels of depressive symptoms. The life-long nature of this association may potentially inform policy agendas aimed at improving urban and suburban living conditions.

## Strengths and limitations of this study

► This is the first paper, to the best of our knowledge, to focus on long-term relationships between childhood urbanicity of residence and late-adulthood depressive outcomes specifically.

► We find evidence pointing towards a link between childhood urbanicity and late-adulthood depressive outcomes and highlight potential pathways for this.

► Due to the administrative structure of our data and the absence of any plausible identifying restrictions for causal effects, we are unable to make strong claims regarding causality.

## INTRODUCTION

The increase in prevalence of mental health disorders in the late 20th and early 21st century, along with increased urban dwelling, raises the question of interplay between these two phenomena. While a substantial body of literature, outlined below, examines links between the lived environment and mental health outcomes at various points over the life course, a potential long-term relationship between childhood urbanicity and late-life depression remains less substantially researched.

A WHO review[1] reports that an estimated 300 million people suffer from depression globally. Associations with further health conditions and wider adverse socioeconomic outcomes are well explored, with one meta-analysis[2] evidencing the multiplicity of social costs of major depression, such as reduced educational outcomes, adverse labour market and health outcomes, and increases in other health-damaging behaviours.

As of 2014, 54% of the world's population resides in urban areas, with this figure having risen from 30% since 1950 and being set to rise

to 66% by 2050.[3] Various pathways have been suggested for links between levels of urbanicity and health outcomes. Vlahov[4] report that the urban social environment can affect individuals' general health via certain features, such as socioeconomic status, crime and violence, presence of marginalised populations and the prevalence of psychological stressors. Diez Roux and Mair[5] further detail potential pathways for links between physical and social aspects of the lived environment and current health outcomes. Compared with living in a rural area, urban dwellers face an elevated amount of environmental stressors on a daily basis, which can lead to long-term degradation of their mental well-being. Further detail of links between city dwelling, quality of life and happiness is provided in ref 6, with aspects of city dwelling such as pollution, monotony of life and the artificiality of the city suggested as being potentially detrimental to quality of life. A Finnish study[7] suggests that this may be due in part to variation in socioeconomic status between rural and urban areas.

Existing literature has generally focused on the relationships between urbanicity and contemporaneous health outcomes at various stages of the life course, with a number of these covering common mental disorders such as depression.[8–13] While the authors caution against drawing causal conclusions, in a broad meta-analysis of the link between urbanicity and mental health disorders, Penkalla and Kohler[14] find links between the degree of urbanicity and mood and anxiety disorders, psychosis and substance use disorders. The largest single study,[15] using administrative data collected on the entire Swedish population of 4.4 million people aged 25–64, finds that those in the top quintile by urbanicity have a 68%–77% higher risk of developing psychosis and a 12%–20% higher risk of developing depression than those in the lowest, with these relationships being stronger for women than for men. Oliver[16] discusses potential links between current suburban and urban living and adverse mental health outcomes.

The relationship between childhood urbanicity and depression later in life remains a topic substantially less explored in existing research. While some evidence[17 18] points to relationships between early life urban dwelling and impacts on brain areas associated with negative affect, we believe there exists only one study[19] that has attempted to carry out preliminary investigations into longer term and later-life relationships that directly use measures of depression. Literature on the life course perspective[20–23] suggests, however, theoretical roles for the impact of conditions throughout an individual's life (particularly of early life conditions) on subsequent health, with this impact likely to vary in particular by gender (as evidenced by theoretical and empirical reviews[24 25]). Various pathways have been proposed to explain the incidence of depression specifically over the life cycle, with Rudenstine[26] emphasising a role for the impact of early life circumstances.

Previous studies examining longer arm relationships between urbanicity and mental ill health suggest empirical support for these theoretical pathways. One study employing the records of over 2.3 million people born in Sweden between 1967 and 1989[27] finds that the population density of area of residence at age 15 is predictive of subsequent schizophrenia. Kim *et al*[19] use the principal areas that participants lived in during early life (up to age 20) and midlife (21–60) to investigate the relationship between the urbanicity of residence at such ages and depression later in life. The study finds links between current urbanicity and current depressive outcomes, and a greater prevalence of depression among women. Individuals who lived their entire life in rural areas are found to have the lowest rates of depression, with similar rates of depression observed among individuals who grew up in rural areas and subsequently lived in urban areas, and those who spent their entire life in urban areas. Lederbogen *et al*,[17] using MRI to examine brain activity, present evidence of relationships between urban dwelling in childhood and social stress processing. A further study[28] uses German and Chinese data to investigate links between childhood urbanicity and emotional characteristics later in (mostly young) adulthood. These results, drawn from relatively small samples (324 individuals in Germany and 713 in China) provide some evidence suggestive of a relationship with negative emotional traits for Chinese women alone, among whom *lower* symptoms of 'fear' are exhibited for those who lived in urban settings in childhood.

In this paper, we seek to investigate specifically the relationship between early-life urban living and the presence of depression in late adulthood. We employ a cross-country European dataset of individuals aged 50 and above in order to investigate such a relationship, using individuals' self-reports of childhood circumstances, and depressive symptoms as measured by the EURO-D questionnaire. The Methods section introduces our data source and variables used in our study. The Results section discusses the results of our study and its implications. The Discussion section concludes.

## METHODS
### Data and sample
For the purpose of our study, we employed two waves of the Survey of Health, Ageing and Retirement in Europe (SHARE), a pan-European survey containing information on health, socioeconomic and social and family networks of individuals aged 50 or older in 13 European countries plus Israel (Israel, however, was not included in our analysis due to its absence from wave 3). SHARE consists of six published waves to date, with individuals tracked across successive surveys. The study design and sampling methods, both in general and for the specific waves employed here, were detailed by the study's authors in several publications.[29–34] We employed the second contemporaneous wave of the study (collected in 2006–2007) and SHARELIFE, the third wave of the survey (collected in 2008–2009), which focuses on the self-reported life

**Table 1** Descriptive statistics (N=20 400)

| Name | All | All | Mean/%<br>Men | Mean/%<br>Women |
|---|---|---|---|---|
| Individual characteristics | | | (n=9029) | (n=11 371) |
| Male (n, %) | 9029 | *44.26* | N/A | N/A |
| Age (mean, SD) | 64.23 | *9.43* | 64.34 | 64.15 |
| Urbanicity of current residence (mean, SD) | 3.33 | *1.41* | 3.35 | 3.31 |
| Urbanicity of childhood residence (mean, SD) | 3.70 | *1.46* | 3.67 | 3.71 |
| Equivalised household income (000s) (mean, SD) | 21.57 | *30.69* | 23.36 | 20.16*** |
| Health status | | | | |
| Depression (EURO-D) (mean, SD) | 2.18 | *2.15* | 1.68 | 2.58*** |
| EURO-D score 4+ (n, %) | 4666 | *22.87* | *14.67* | *29.38\*\*\** |
| Childhood socioeconomic status (SES) | | | | |
| Number of rooms (mean, SD) | 3.72 | *1.79* | 3.73 | 3.71 |
| Number of facilities (mean, SD) | 2.28 | *1.48* | 2.28 | 2.28 |
| Number of books (1=few to 5=>200 books) (mean, SD) | 2.13 | *1.21* | 2.11 | 2.14 |
| Occupation of main breadwinner (ranked) (mean, SD) | 2.90 | *0.80* | 2.90 | 2.91 |
| Childhood characteristics | | | | |
| Math performance (1=low to 5=high) (mean, SD) | 3.30 | *0.89* | 3.39 | 3.23** |
| Language performance (1=low to 5=high) (mean, SD) | 3.33 | *0.87* | 3.24 | 3.41** |
| Self-reported health (1=poor to 5=excellent) (mean, SD) | 3.97 | *1.00* | 4.04 | 3.91** |
| Parent heavy drinker (n, %) | 1658 | *8.13* | *8.09* | *8.16* |
| Biological father present (n, %) | 18 582 | *91.09* | *91.15* | *91.04* |
| Biological mother present (n, %) | 19 639 | *96.27* | *3.41* | *3.98\** |
| Stepfather present (n, %) | 441 | *2.16* | *2.10* | *2.21* |
| Stepmother present (n, %) | 322 | *1.58* | *1.46* | *1.67* |

Mean (for non-binary variables) or number (for binary variables) is given in normal font and standard deviation (for non-binary variables) or percentage (for binary variables) is given in italics.
*Asterisks indicate statistical significance.
N/A, not applicable.

histories of respondents. In total, 25 341 individuals appeared in both datasets. After a listwise removal of individuals with missing data for individual variables, our final sample stood at 20 400 individuals (9029 men and 11 371 women; we also used a larger sample (consisting of 24 854 individuals) for our full information maximum likelihood (FIML) analysis containing all individuals in both waves of the data with a recorded EURO-D score) and 13 548 individuals (6154 men and 7394 women) when control was made for current urbanicity. A summary of item missingness is shown in online supplementary table A1. Table 1 shows the summary statistics of the final sample used in the study, broken down by gender and by childhood urbanicity.

### Variables
The main dependent variable used in this study was depression, with symptoms measured by the EURO-D scale included in the SHARE questionnaire. The EURO-D is a symptoms scale for depression and has been validated in a cross-European study.[35] The scale covers 12

symptoms: depressed mood, pessimism, suicidality, guilt, sleep, interest, irritability, appetite, fatigue, concentration, enjoyment and tearfulness. The final EURO-D score (between 0 and 12) is calculated from the total number of these symptoms present, with a score of 4 or more found to be an optimal cut-off point for the prediction of depression.[35 36] In addition to a set of models that treat this EURO-D score as the dependent variable in ordinary least squares (OLS) regressions, in a second set of models, we transformed this variable into a dichotomous variable that was equal to 1 for EURO-D scores of 4 and higher, and 0 for scores strictly below 4.

The main independent variable of interest in our study is the urbanicity of the respondents' childhood homes. In the retrospective SHARELIFE questionnaire, respondents provided information on each dwelling they had lived in since birth, providing the exact year in which they moved and the type of area in which each dwelling was located. This area type is coded in SHARELIFE with values of 1 ('a big city'), 2 ('the suburbs or outskirts of a big city'), 3

('a large town'), 4 ('a small town') and 5 ('a rural area or village'). A previous study investigating the recall of other childhood conditions in SHARE[37] presents evidence that individuals recall these in general accurately.

Durations, measured in years, between moving into and out of each dwelling were computed to determine the length that respondents lived in each of these. We then identified the home that respondents lived in for the longest time between birth and the age of 15, and the corresponding area type of that dwelling. In our base case, we included this variable as is, after the exclusion of cases for non-response.

## Statistical analysis

We outline here the various statistical methods employed. We estimated our models primarily as a series of OLS and logistic regressions, the former being used for regressions using the EURO-D score as the dependent variable and the latter for those using a dichotomised EURO-D variable equal to 1 for scores of 4 or more, and to 0 for scores strictly fewer than 4. We also included an FIML specification that allowed us to estimate parameters using all available data in the sample, under the assumption that observations are missing at random,[38 39] as a final robustness check.

We included basic demographic controls of gender and age–period–cohort in all models and allowed for country fixed effects. In summarising the varying social, marital and economic positions of men and women and the potential tendencies of each to respond to adverse life events in different ways, existing literature[24 40] suggests there may be differences both in overall prevalence of depression by gender and also in relationships between explanatory factors and such mental health outcomes. Furthermore, while all respondents in the SHARELIFE survey are aged above 50, existing literature points to a role for age as a relevant control variable even among this group.[41 42] Although interpretation of the variation explained by age variables in studies such as this is plagued by problems in separately identifying age–period–cohort effects—that is, due to collinearity, identifying these three effects independently is impossible without resorting to additional, untestable, assumptions[43]—we include these variables purely as controls for the combination of all three potential such effects. While for brevity we subsequently referred to these variables as 'age' controls, we remained agnostic as to the actual source of variation by age–period–cohort and merely attempted to control for any such relationship. In an initial parsimonious model, we controlled only for these basic demographic factors.

In further models, we also included additional control variables related to childhood circumstances and characteristics, and contemporaneous log income. (We implemented this by dividing gross household income by the square root of household size, as per the current method employed by recent Organisation for Economic Co-operation and Development publications.[44 45]) Latham[46] reported that childhood disability is significantly associated with later life depression, a relationship found to be of an indirect nature through social and health factors such as midlife physical health. To control for such an effect in our model, we introduced self-reported childhood health status, a score between 1 (poor) and 5 (excellent). We also sought to control for childhood socioeconomic circumstances and, to this end, followed a validated method[37] employed in other papers[47 48] employing SHARE, a principal component score, created from a combination of measures of childhood (age 10) socioeconomic status included in SHARELIFE. Online supplementary table A2 in the online supplementary appendix provides the results of our principal component analysis. We used only component 1 to construct the score used in our index of childhood socioeconomic status (SES), which we divided into deciles in our model. We also added parental presence (split by step-parent and biological parent) in the childhood home at the age of 10, self-perceived and self-recalled ability at age 10 in mathematics and literacy compared with the average and self-reported parental alcohol abuse.

In a final model, we sought to consider the relationship between depression in late adulthood and childhood urbanicity once control was made for current urbanicity at the time at which the SHARE interview was conducted. To this end, we included a variable for current urbanicity coded in an identical way to that in which we coded childhood urbanicity, as earlier described.

We also presented alternative specifications, changing the dependent variable to the urbanicity of area of residence when aged between 0 and 5, 6 and 10, and 11 and 15 years.

## RESULTS

Out of the 20 400 individuals included in our main sample, 23% were recorded as having a EURO-D score of 4 or above, with an average score of 2.18. We found that women (29%) were almost twice as likely as men (15%) to exhibit a EURO-D score of 4 or above, and this score for women (2.58) was just under 1 point higher on average than for men (1.68).

Our results suggested that the urbanicity of childhood residence, rather than of current residence, was of primary importance in the relationship between the lived environment and depression. These results also suggested that such a relationship varied by gender, with an F-test of the interactions between gender and childhood urbanicity categories being rejected at p<0.0001. A matrix of individuals' urbanicity of residence in childhood and currently is presented in online supplementary table A3.

Our results suggested a complex relationship between the level of childhood urbanicity and later life depression. A parsimonious pooled regression (column 1, table 2) including both men and women suggested that, compared with individuals living in rural areas or villages, only individuals spending their childhoods in the suburbs of big cities were observed to have significantly worse outcomes

**TABLE 2** Association between childhood urbanicity and late-adulthood EURO-D score (ordinary least squares coefficients)

| | Full sample | | | Men | Men | Women | Women |
|---|---|---|---|---|---|---|---|
| | EURO-D | EURO-D | EURO-D | EURO-D | EURO-D | EURO-D | EURO-D |
| Dependent variable | 1 | 2 | 3 | 4 | 5 | 6 | 7 |
| Age 0–15 living environment (omitted: rural area or village) | | | | | | | |
| Big city | 0.0474 | 0.0978** | 0.117** | 0.104* | 0.0755 | 0.0937 | 0.151* |
| | (0.0432) | (0.0440) | (0.0566) | (0.0582) | (0.0740) | (0.0641) | (0.0836) |
| Suburbs or outskirts of a big city | 0.117** | 0.153*** | 0.156** | 0.0563 | 0.0355 | 0.223*** | 0.253** |
| | (0.0560) | (0.0562) | (0.0704) | (0.0751) | (0.0913) | (0.0812) | (0.105) |
| Large town | 0.0208 | 0.0760* | 0.0977* | 0.0653 | 0.0812 | 0.0834 | 0.124 |
| | (0.0433) | (0.0436) | (0.0558) | (0.0583) | (0.0726) | (0.0631) | (0.0826) |
| Small town | −0.0275 | 0.0136 | 0.0153 | 0.0818 | 0.114* | −0.0471 | −0.0791 |
| | (0.0412) | (0.0410) | (0.0518) | (0.0552) | (0.0683) | (0.0590) | (0.0759) |
| Male | −0.887*** | −0.846*** | −0.864*** | | | | |
| | (0.0286) | (0.0287) | (0.0355) | | | | |
| Current living environment (omitted: rural area or village) | | | | | | | |
| Big city | | | −0.0850 | | −0.0755 | | −0.0634 |
| | | | (0.0603) | | (0.0800) | | (0.0883) |
| Suburbs or outskirts of a big city | | | −0.0528 | | 0.0267 | | −0.115 |
| | | | (0.0588) | | (0.0759) | | (0.0879) |
| Large town | | | 0.0299 | | −0.0670 | | 0.136* |
| | | | (0.0541) | | (0.0715) | | (0.0792) |
| Small town | | | −0.0840 | | −0.0868 | | −0.0669 |
| | | | (0.0514) | | (0.0671) | | (0.0760) |
| Additional childhood controls and contemporaneous income | No | Yes | Yes | Yes | Yes | Yes | Yes |
| Observations | 20 400 | 20 400 | 13 548 | 9029 | 6154 | 11 371 | 7394 |

Constant was not reported.

All models contain controls for age–period–cohort and country fixed effects. Childhood controls consist of self-reported and self-recalled health status, a principal component score for deprivation, self-perceived and self-recalled numeracy and literacy at age 10, and self-reported parental alcohol abuse.

Column 1 displays the estimation results from regressing the EURO-D depression score on a set of living conditions when the respondent was aged 0–15. Column 2 adds various childhood and contemporaneous controls, and column 3 controls for current living conditions. Columns 4–7 repeat the specifications of columns 2 and 3 for men and women separately, respectively.

Asterisks indicate statistical significance.

in late adulthood. Our model with childhood controls (column 2, table 2) suggested that individuals growing up in big cities were observed to have EURO-D scores of 0.0978 (p<0.05), on average worse than those growing up in rural areas or villages. In the same model, individuals growing up in the suburbs of big cities have worse still depressive outcomes as measured by the EURO-D score, with this score being on average 0.153 (p<0.01) higher for such individuals. Individuals growing up in large towns were estimated to have a EURO-D score in adulthood on average 0.0760 (p<0.10) higher than those growing up in rural areas. These results were robust to respecification of the model with our age variable modified to, variously, the type of area lived for longest between the ages of 0 and 5, 6 and 10, and 11 and 15 (online supplementary table A4). (We also ran our analysis incorporating both

marital status and self-reported parental mental health conditions. In each case, there was no change in statistical significance and no substantive change in coefficients associated with area type in childhood. Full results are available on request.)

When broken down by gender, these relationships across our models appeared to differ substantially. Much stronger results were observed for women than for men. While only the coefficient for having spent childhood in a big city reached conventional levels of significance for men (column 4, table 2), this relationship was only observed for women for those who grew up in the suburbs of a big city (column 6, table 2). Men with childhoods spent in a big city were estimated to have a EURO-D score of on average 0.104 (p<0.10) higher than those who grew up in rural areas or villages, while women with suburban

**Table 3** Association between childhood urbanicity and late-adulthood depression incidence (logit, ORs)

| | Full sample | | | Men | Men | Women | Women |
|---|---|---|---|---|---|---|---|
| | 1 | 2 | 3 | 4 | 5 | 6 | 7 |
| Age 0–15 living environment (omitted: rural area or village) | | | | | | | |
| Big city | 1.045 | 1.121** | 1.100 | 1.129 | 1.021 | 1.117 | 1.147 |
| | (0.0566) | (0.0634) | (0.0797) | (0.109) | (0.127) | (0.0783) | (0.103) |
| Suburbs or outskirts of a big city | 1.176** | 1.235*** | 1.217** | 1.039 | 0.990 | 1.334*** | 1.350*** |
| | (0.0810) | (0.0876) | (0.108) | (0.132) | (0.152) | (0.115) | (0.148) |
| Large town | 1.036 | 1.109* | 1.087 | 1.084 | 1.047 | 1.120* | 1.116 |
| | (0.0556) | (0.0615) | (0.0765) | (0.105) | (0.127) | (0.0761) | (0.0976) |
| Small town | 0.996 | 1.045 | 1.057 | 1.106 | 1.143 | 1.014 | 1.007 |
| | (0.0504) | (0.0540) | (0.0685) | (0.0993) | (0.125) | (0.0643) | (0.0812) |
| Male | 0.399*** | 0.410*** | 0.416*** | | | | |
| | (0.0149) | (0.0157) | (0.0193) | | | | |
| Current living environment (omitted: rural area or village) | | | | | | | |
| Big city | | | 0.947 | | 0.927 | | 0.980 |
| | | | (0.0716) | | (0.121) | | (0.0916) |
| Suburbs or outskirts of a big city | | | 0.984 | | 1.071 | | 0.945 |
| | | | (0.0737) | | (0.132) | | (0.0890) |
| Large town | | | 1.052 | | 0.915 | | 1.155* |
| | | | (0.0705) | | (0.106) | | (0.0961) |
| Small town | | | 0.871** | | 0.825* | | 0.906 |
| | | | (0.0565) | | (0.0908) | | (0.0732) |
| Additional childhood controls and contemporaneous income | No | Yes | Yes | Yes | Yes | Yes | Yes |
| Observations | 20 400 | 20 400 | 13 548 | 9029 | 6154 | 11 371 | 7394 |

Constant was not reported.

All models contain controls for age–period–cohort and country fixed effects. Childhood controls consist of self-reported and self-recalled health status, a principal component score for deprivation, self-perceived and self-recalled numeracy and literacy at age 10, and self-reported parental alcohol abuse.

Column 1 displays the estimation results from regressing the EURO-D depression score on a set of living conditions when the respondent was aged 0–15. Column 2 adds various childhood and contemporaneous controls, and column 3 controls for current living conditions. Columns 4–7 repeat the specifications of columns 2 and 3 for men and women separately, respectively.

Asterisks indicate statistical significance.

childhoods were observed to have a EURO-D score of on average 0.223 (p<0.01) higher than those who grew up in rural areas or villages.

While these may appear to represent small increases, it is instructive to also consider the results of a logit model with presence of a EURO-D score of 4 or more as dependent variable, with these results presented in table 3 (ORs) and table 4 (average marginal effects). Such a model again suggested that suburban childhoods were associated with worse depressive outcomes for women, with an average marginal effect estimated for women of 5.58 percentage points in our preferred model.

While we confirm a role for childhood urbanicity in explaining adult urbanicity (with an F-test of the relevance of such variables in an ordered logit model with adulthood urbanicity as the dependent variable being rejected at p<0.0001), the addition of this as an explanatory variable (columns 3, 5 and 7; table 2) does not substantially affect our estimated relationship between urbanicity of childhood residence and the individual's EURO-D score in late adulthood. (Although there was a reduction in statistical significance (indeed a loss of significance for our big city point estimate), it is worth noting that the addition of current urbanicity reduced our sample size by approximately one-third. None of our point estimates were substantially affected by the addition of current urbanicity.)

## DISCUSSION

These results are suggestive of a long–term relationship, persisting beyond the age of 50, between levels of childhood urbanicity and depression. The presence of such a relationship is robust to several model specifications and is most commonly observed for those who grew up in suburban areas. In contrast to existing literature, we

**Table 4** Association between childhood urbanicity and late-adulthood depression incidence (logit, average marginal effects)

| | Full Sample | | | Men | Men | Women | Women |
|---|---|---|---|---|---|---|---|
| | (1) | (2) | (3) | (4) | (5) | (6) | (7) |
| *Age 0–15 living environment (omitted: rural area or village)* | | | | | | | |
| Big city | 0.00717 | 0.0181** | 0.0151 | 0.0144 | 0.00240 | 0.0208 | 0.0261 |
| | (0.00878) | (0.00903) | (0.0116) | (0.0117) | (0.0147) | (0.0133) | (0.0173) |
| Suburbs or outskirts of a big city | 0.0268** | 0.0340*** | 0.0319** | 0.00443 | −0.00118 | 0.0558*** | 0.0586*** |
| | (0.0117) | (0.0118) | (0.0147) | (0.0148) | (0.0179) | (0.0172) | (0.0220) |
| Large town | 0.00569 | 0.0162* | 0.0132 | 0.00944 | 0.00546 | 0.0212* | 0.0209 |
| | (0.00868) | (0.00881) | (0.0112) | (0.0115) | (0.0144) | (0.0129) | (0.0167) |
| Small town | −0.000625 | 0.00683 | 0.00875 | 0.0118 | 0.0163 | 0.00253 | 0.00129 |
| | (0.00806) | (0.00806) | (0.0102) | (0.0107) | (0.0136) | (0.0117) | (0.0150) |
| Male | −0.148*** | −0.141*** | −0.140*** | | | | |
| | (0.00580) | (0.00583) | (0.00714) | | | | |
| *Current living environment (omitted: rural area or village)* | | | | | | | |
| Big city | | | −0.00870 | | −0.00923 | | −0.00388 |
| | | | (0.0120) | | (0.0157) | | (0.0178) |
| Suburbs or outskirts of a big city | | | −0.00255 | | 0.00866 | | −0.0108 |
| | | | (0.0120) | | (0.0157) | | (0.0178) |
| Large town | | | 0.00823 | | −0.0107 | | 0.0282* |
| | | | (0.0109) | | (0.0139) | | (0.0163) |
| Small town | | | −0.0216** | | −0.0225* | | −0.0185 |
| | | | (0.0101) | | (0.0128) | | (0.0151) |
| Additional childhood controls and contemporaneous income | No | Yes | Yes | Yes | Yes | Yes | Yes |
| Observations | 20 400 | 20 400 | 13 548 | 9029 | 6154 | 11 371 | 7394 |

Constant was not reported.

All models contain controls for age–period–cohort and country fixed effects. Childhood controls consist of self-reported and self-recalled health status, a principal component score for deprivation, self-perceived and self-recalled numeracy and literacy at age 10, and self-reported parental alcohol abuse.

Column 1 displays the estimation results from regressing the EURO-D depression score on a set of living conditions when the respondent was aged 0–15. Column 2 adds various childhood and contemporaneous controls, and column 3 controls for current living conditions. Columns 4–7 repeat the specifications of columns 2 and 3 for men and women separately, respectively.

Asterisks indicate statistical significance.

find that increasing levels of contemporaneous urbanicity generally have a small and non-significant negative relationship with both EURO-D score (in our linear regressions) and presence of a EURO-D score of 4 or more (in our logistic regressions). Furthermore, the inclusion of this additional variable does not substantially impact on the magnitude of our parameter estimates of the relationship between childhood urbanicity and depressive outcomes. These small and statistically insignificant parameter estimates suggest little evidence of a relationship between current urbanicity and the presence of depression in our sample.

With the continued trend of rapid urbanisation and the rising prominence of mental health conditions,[1 3] the investigation of links between childhood urbanicity and mental health outcomes is required to help policy makers better understand and design policies to cater to such growing needs. Our study follows in the footsteps of existing literature on urbanicity and mental health but goes on to specifically look at the relationship of urbanicity and depression from a long-term standpoint, as well as being, to our knowledge, the first of this type to consider separate relationships by gender. In contrast to existing studies in the field, our study employs a cross-national sample with a comprehensive and wide range of retrospective data.

In line with literature in the life course perspective emphasising the importance of early life circumstances, we find that spending one's childhood in an urban or suburban area has a positive and significant relationship with the presence of depression in later life adulthood when a pooled regression is run for the entire sample. This relationship persists even after controlling for other socioeconomic, demographic and environmental factors. This relationship appears to be particularly strong for women. Individuals who spend the longest time in their

childhood in a suburban home face an average increase in probability of 3.4 percentage points (5.6 points for women) of reporting four or more depressive symptoms on the EURO-D scale during later life adulthood, a relationship that persists even when control is made for current urbanicity. It is important to acknowledge that the nature of SHARE is such that going beyond establishing correlations in our data is not possible. While we are unable to establish any identifying restrictions for such causal effects in our dataset, the findings here are striking and establish potential pathways for future research. Although SHARE's rich retrospective information, even if based on recall data, provides a rare opportunity to investigate long-term relationships such as this, the use of linked administrative long panel datasets would be preferable. Such datasets would contain more comprehensive information about specific areas of residence and would help to overcome one limitation of our study: that of treating 'big city' as one homogeneous item, borne out of our inability to account for the fact that the experience of city life varies both within and between cities. Future research could seek to overcome some of these inherent limitations of our approach, in seeking to establish causal effects, as well as to establish theoretical pathways for such effects more clearly.

**Acknowledgements** This paper uses data from SHARE waves 2 and 3 (SHARELIFE) (DOIs: 10.6103/SHARE.w2.610 and 10.6103/SHARE.w3.610); see Börsch-Supan *et al* (2013) for methodological details. The SHARE data collection has been primarily funded by the European Commission through FP5 (QLK6-CT-2001-00360), FP6 (SHARE-I3: RII-CT-2006-062193, COMPARE: CIT5-CT-2005-028857 and SHARELIFE: CIT4-CT-2006-028812) and FP7 (SHARE-PREP: N°211909, SHARE-LEAP: N°227822 and SHARE M4: N°261982). Additional funding from the German Ministry of Education and Research, the Max Planck Society for the Advancement of Science, the US National Institute on Aging (U01_AG09740-13S2, P01_AG005842, P01_AG08291, P30_AG12815, R21_AG025169, Y1-AG-4553-01, IAG_BSR06-11, OGHA_04-064 and HHSN271201300071C) and various national funding sources is gratefully acknowledged (see http://www.share-project.org/home0.html). We gratefully acknowledge the insightful comments of referees.

**Contributors** SL carried out the initial empirical estimation and wrote the first draft as a postgraduate thesis under the supervision of DH and JM. DH carried out the re-estimation and estimation of additional model specifications and substantially redrafted the paper. JM and SL made further redrafting contributions to the paper. All authors approved the final draft.

**Funding** This project was supported by the Investeringsagenda of the University of Groningen.

**Competing interests** None declared.

**Patient consent for publication** Not required.

**Ethics approval** An ethics statement related to SHARE/SHARELIFE, the only data on which we rely for this paper, is available online (http://www.shareproject.org/fileadmin/pdf_documentation/SHARE_ethics_approvals.pdf). No public or patient involvement beyond this took place.

**Provenance and peer review** Not commissioned; externally peer reviewed.

**Data availability statement** Data are available in a public, open access repository.

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
