## [Reviewer comments · BMJ Open]

ARTICLE DETAILS

TITLE (PROVISIONAL)	The Relationship Between Early Life Urbanicity and Depression in Late-Adulthood: Evidence from the Survey of Health, Ageing and Retirement in Europe
AUTHORS	Howdon, Daniel; Mierau, Jochen; Liew, Samuel

VERSION 1 – REVIEW

REVIEWER	Adam J Okulicz-Kozaryn Rutgers
REVIEW RETURNED	08-Dec-2018

GENERAL COMMENTS	really good research area and approach; you need to cite: Lederbogen, Florian et al City living and urban upbringing affect neural social stress processing in humans; nice description of dataset; what is missing is the mechanism or causal path--why would growing up in a city make one depressed later in life--for a recent overview of the literature see: https://www.palgrave.com/us/book/9781137436320 specifically there is some sociological classical literature on this, eg see Wirth, Toennies, Simmel; per modeling: how about having country dummies?
--

REVIEWER	Alicia Colvin University of Pittsburgh, USA
REVIEW RETURNED	24-Jan-2019

GENERAL COMMENTS	Overall, this is a very nicely organized and well-written manuscript. I have the following specific comments for the authors: 1. Methods section: there is not statement on participant consent or ethics approval. The authors reference the SHARE study but should still provide a statement in their manuscript addressing this. 2. Page 8, last paragraph, last sentence of the Variables section: Please clarify what is meant by "we include this variable as is, after cleaning"? What did "cleaning" entail? How did the variable change as a result? 3. Page 9: Reference 36 is cited in the section discussing age and cohort effects, yet this reference is for a paper examining parental smoking and adolescent behavior. This I believe is an error. 4. Why was marital status not included as a basic demographic variable?
--

	5. Tables: In the proof Table 4 seems to appear twice, and there is no Table 5 (Table 5 is listed to be inserted into the text, but perhaps there is not supposed to be a Table 5?). 6. It would be helpful for the authors to include a footnote in the tables briefly listing the additional childhood control variables that are included so the reader does not have to refer back to the text and so the tables can stand on their own. 7. Page 11, Results, line 47: The authors state that in the pooled parsimonious regression, individuals who spent childhood living in cities or suburbs have significantly worse outcomes. The table supports this statement for suburbs but not for cities. The estimate for cities is relatively small and does not reach statistical significance. 8. Page 12, Results, line 3: I believe this should read "individual living in LARGE towns are estimated to have a EURO-D score in adulthood on average 0.0760...". The text currently reads "small towns", and this does not match the table. 9. Page 12, Results, line 52: The authors state that adding adulthood urbanicity does not substantially affect estimated relationship between childhood urbanicity and EURO-D in adulthood. But for men, when adulthood urbanicity is added, the estimate for childhood big city decreases and is no longer significant. Also, the estimate for small town increases and becomes significant. For women, the estimate for childhood big city increases and becomes significant. 10. The Discussion could be improved by some more detailed discussion of why childhood suburban residence seemed to be so significant (even more so than city) and the differences found between men and women - some comparisons with the existing literature and why you think your results may be similar/different. Also a bit more discussion of the limitations, particularly the retrospective recall of childhood data, would be helpful. 11. Did SHARE collect information of family history of depression/mental health issues? If it did, this would be an important variable to examine, and if not, an important issue (lack of ability to look at family history of depression) to address in the Discussion.
--	---

REVIEWER	Carla Bezold NA, USA
REVIEW RETURNED	30-Jan-2019

GENERAL COMMENTS	The results of this paper do not support the conclusions drawn by the authors. The methods are at times unclear and/or unjustified, or there is insufficient data provided to evaluate the approach used. The paper would benefit from major revisions including potentially an alternative model for the primary analysis, as OLS regression does not seem appropriate. Introduction: Page 4, Lines 33-55: Please specify more precisely the sorts of associations observed in the papers you are citing, particularly for the Diez Roux paper
---

	Page 6, Line 8: This paragraph is about urbanicity and later life outcomes but you report concurrent associations. Please focus this section on the most relevant results Page 6, Lines 20-31: It is unclear how this study fits with the larger argument. In general this paragraph would benefit from a more clear argument. Methods: Page 7, Line 32: Please specify what “control for contemporary urbanicity” means Page 8, Line 5: Is OLS regression appropriate for a variable with an outcome range of 12, where most scores are clustered in the low end of the range? Please justify the use of this model or present a different, more appropriate, regression model. Page 8, Line 25: Have any efforts been made to validate this self-reported scale of urbanicity? Page 9: How did you control for age-period-cohort effects? Page 9: contemporaneous income could conceivably be on the causal path if I am thinking about this right. Please clarify how and when income was measured. You discuss results of an F-test a number of times but it is not clear in the methods when and how this was used Results: Page 11: The sentence beginning “They also suggest that such a relationship...” Page 12, Line 42: Please specify how you calculated the average marginal effect Discussion: Overall most of the results are null. The authors focus on a handful of results that are statistically significant, but the majority are null. Please temper the discussion to clarify this. The authors overstate their findings somewhat in the current draft. Page 14, Line 7: What other environmental factors were considered? Tables Table 1: Please report proportions as N(%) rather than mean (SD) Table 2: Please include footnotes in the table explaining what each column indicates. It is a little hard to follow in the current set up
--	--

VERSION 1 – AUTHOR RESPONSE

Reviewer: 1

Reviewer Name: Adam J Okulicz-Kozaryn

Institution and Country: Rutgers

Please state any competing interests or state ‘None declared’: none

Please leave your comments for the authors below really good research area and approach; you need to cite:

Lederbogen, Florian et al City living and urban upbringing affect neural social stress processing in humans; nice description of dataset

We have now cited this paper.

what is missing is the mechanism or causal path--why would growing up in a city make one depressed later in life--for a recent overview of the literature see:
<https://www.palgrave.com/us/book/9781137436320>
specifically there is some sociological classical literature on this, eg see Wirth, Toennies, Simmel;

We have now partly rewritten our introduction to provide greater detail on potential causal pathways.

per modeling: how about having country dummies?

We include country dummies in all estimated models. This is currently detailed in paragraph two of our Statistical analysis section. There is additionally now a footnote to each table stating this – if preferred we could add this back to our table.

Reviewer: 2

Reviewer Name: Alicia Colvin

Institution and Country: University of Pittsburgh, USA Please state any competing interests or state 'None declared': None declared

Please leave your comments for the authors below Overall, this is a very nicely organized and well-written manuscript. I have the following specific comments for the authors:

1. Methods section: there is not statement on participant consent or ethics approval. The authors reference the SHARE study but should still provide a statement in their manuscript addressing this.

We have now added an in-text citation for this.

2. Page 8, last paragraph, last sentence of the Variables section: Please clarify what is meant by "we include this variable as is, after cleaning"? What did "cleaning" entail? How did the variable change as a result?

We agree that this was poorly worded, and now clarify that this was simply removal of cases due to non-response.

3. Page 9: Reference 36 is cited in the section discussing age and cohort effects, yet this reference is for a paper examining parental smoking and adolescent behavior. This I believe is an error.

This has now been amended to ensure papers are cited at their correct positions.

4. Why was marital status not included as a basic demographic variable?

We have carried out re-estimation with marital status included as a variable. This does not substantially change our estimated coefficients and does not affect statistical significance. A sample comparison (that in column 2 of table 1) is provided below:

areatype_0to15	Included	Excluded	significance (both)
1	0.084034	0.097844	**
2	0.140635	0.152623	***
3	0.077369	0.075961	*
4	0.009387	0.013567	

5. Tables: In the proof Table 4 seems to appear twice, and there is no Table 5 (Table 5 is listed to be inserted into the text, but perhaps there is not supposed to be a Table 5?).

We are grateful to the reviewer for spotting this error. Table 4 was indeed previously duplicated and Table 5 was referred to in our typesetting notes in error.

6. It would be helpful for the authors to include a footnote in the tables briefly listing the additional childhood control variables that are included so the reader does not have to refer back to the text and so the tables can stand on their own.

This has now been added to all results tables.

7. Page 11, Results, line 47: The authors state that in the pooled parsimonious regression, individuals who spent childhood living in cities or suburbs have significantly worse outcomes. The table supports this statement for suburbs but not for cities. The estimate for cities is relatively small and does not reach statistical significance.

We are grateful to the reviewer for identifying this error and have corrected this.

8. Page 12, Results, line 3: I believe this should read "individual living in LARGE towns are estimated to have a EURO-D score in adulthood on average 0.0760...". The text currently reads "small towns", and this does not match the table.

We are again grateful to the reviewer for identifying this error and have corrected this.

9. Page 12, Results, line 52: The authors state that adding adulthood urbanicity does not substantially affect estimated relationship between childhood urbanicity and EURO-D in adulthood. But for men, when adulthood urbanicity is added, the estimate for childhood big city decreases and is no longer significant. Also, the estimate for small town increases and becomes significant. For women, the estimate for childhood big city increases and becomes significant.

It is worth noting that the addition of current urbanicity causes a reduction in sample size of one third. While there is a loss of statistical significance in this single case, none of the point estimates are substantially amended. We have added a footnote to this effect but think that practical significance is at least as important as statistical significance, especially when the obtained coefficients are in line with existing point estimates.

10. The Discussion could be improved by some more detailed discussion of why childhood suburban residence seemed to be so significant (even more so than city) and the differences found between men and women - some comparisons with the existing literature and why you think your results may be similar/different. Also a bit more discussion of the limitations, particularly the retrospective recall of childhood data, would be helpful.

We have added more detail on our confidence on retrospective recall of childhood data. We have added greater acknowledgement that we do not seek to claim causality in our results. Our introduction, as per our response to reviewer 1, now features greater detail regarding pathways through which this relationship could work. As we state, we are not aware of other research estimating a comparable relationship.

11. Did SHARE collect information of family history of depression/mental health issues? If it did, this would be an important variable to examine, and if not, an important issue (lack of ability to look at family history of depression) to address in the Discussion.

We are again grateful to the reviewer for this suggestion and have carried out re-estimation based on this. Wave 3 does indeed collect data on self-recall of parental mental health conditions. Only 2% of respondents answer yes to this question. We have however again re-estimated models with this variable included. Again it does not substantially cause any change to estimated coefficients or significance levels. As in our response to question 4, we present estimated coefficients obtained in re-estimation.

	areatype_0to15 Included	Excluded	significance (both)
1	0.094074	0.097844	**
2	0.149086	0.152623	***
3	0.075037	0.075961	*
4	0.014398	0.013567	

Reviewer: 3

Reviewer Name: Carla Bezold

Institution and Country: NA, USA

Please state any competing interests or state 'None declared': None declared

Please leave your comments for the authors below The results of this paper do not support the conclusions drawn by the authors. The methods are at times unclear and/or unjustified, or there is insufficient data provided to evaluate the approach used. The paper would benefit from major revisions including potentially an alternative model for the primary analysis, as OLS regression does not seem appropriate.

These comments seem to be further developed upon by the reviewer below and we reply to them in turn.

Introduction:

Page 4, Lines 33-55: Please specify more precisely the sorts of associations observed in the papers you are citing, particularly for the Diez Roux paper Page 6, Line 8: This paragraph is about urbanicity and later life outcomes but you report concurrent associations. Please focus this section on the most relevant results

We have added the word “current” to this sentence. As detailed, however, a longer-term relationship between childhood urbanicity and health outcomes later in life is – to the best of our knowledge and following literature searches – not well-researched. We feel that the inclusion of hypothesised and evidenced pathways for current such relationship is relevant in both offering pointers towards explaining, as well as contextualising, our research.

Page 6, Lines 20-31: It is unclear how this study fits with the larger argument . In general this paragraph would benefit from a more clear argument.

This paragraph has now been rewritten and an additional study added.

Methods:

Page 7, Line 32: Please specify what “control for contemporary urbanicity” means Page 8,

We have now amended the paper to consistently use “current urbanicity”, as further defined on page 10.

Line 5: Is OLS regression appropriate for a variable with an outcome range of 12, where most scores are clustered in the low end of the range? Please justify the use of this model or present a different, more appropriate, regression model.

We present and discuss two types of model: one attempting to identify an average marginal relationship with the EURO-D score, and another attempting to identify changes in the probability of a EURO-D score of 4 or more, which we argue represents presence of depression. For the former type of model, we estimate this average marginal relationship on the EURO-D score and believe that this is appropriate. Our dataset consists of in excess of 20,000 observations, and appeal to the central limit theorem to identify such a marginal relationship given such a sample size would seem appropriate. For the latter type of model, we do not employ OLS but employ logistic regression which can directly accommodate this binary outcome.

Page 8, Line 25: Have any efforts been made to validate this self-reported scale of urbanicity?

We are unaware of any such efforts to validate this scale specifically but now (at the point where we introduce discussion of this measure) cite Havari, E. & Mazzonna, F. Eur J Population (2015) (already cited elsewhere in the paper) and add a sentence regarding its findings that recall of other variables included in SHARE is generally accurate.

Page 9: How did you control for age-period-cohort effects?

Age-period-cohort effects are not & cannot be separately controlled for due to the inherent collinearity problem discussed. To clarify, we now add an additional sentence “While for brevity we subsequently refer to these variables as “age” controls, we remain agnostic as to the actual source of variation by age-period-cohort and merely attempt to control for any such relationship.”

Page 9: contemporaneous income could conceivably be on the causal path if I am thinking about this right. Please clarify how and when income was measured.

Our aim with this paper is not to identify causal effects but to try to wash out any potential confounding variables – some of which will be on the causal path – and see if any residual relationship remains. We completely agree that contemporaneous income is likely to be on the causal path and our inclusion of this variable is to this end. Income is measured contemporaneously – i.e. it is included for the wave featuring current conditions described in our data.

You discuss results of an F-test a number of times but it is not clear in the methods when and how this was used

The F-tests were used to assess the joint significance of including either interaction effects or childhood circumstances. Being a widely used extension of single parameter significance testing, we felt it superfluous to discuss the test in the methods.

Results:

Page 11: The sentence beginning “They also suggest that such a relationship...”

We are slightly unclear as to the reviewer’s query on this point. We have changed the start of this sentence to “These results also suggest...”.

Page 12, Line 42: Please specify how you calculated the average marginal effect

Due to the nonlinearity inherent in logistic regression, marginal effects cannot be straightforwardly calculated from coefficients alone. We calculate average marginal effects for our sample by using the standard procedure of calculating counterfactual calculations for each observation in the sample, and taking the mean of the difference across all individuals.

Discussion:

Overall most of the results are null. The authors focus on a handful of results that are statistically significant, but the majority are null. Please temper the discussion to clarify this. The authors overstate their findings somewhat in the current draft.

We have rewritten the start of our discussion section to reflect this.

Page 14, Line 7: What other environmental factors were considered?

Unfortunately SHARE does not provide a great deal of such detail. We do not attempt to pin down specifics of what aspects of, for instance, different degrees of urbanicity may affect (if indeed a causal relationship exists) depression later in life and acknowledge this in our conclusions (e.g. our comments regarding “treating “big city” as one homogeneous item”). We do not doubt that the picture is more complicated than our data allows and hope the reviewer agrees that our comments acknowledge this.

Tables

Table 1: Please report proportions as N(%) rather than mean (SD)

We have now done this.

Table 2: Please include footnotes in the table explaining what each column indicates. It is a little hard to follow in the current set up

We have now added a footnote to each table detailing this.

VERSION 2 – REVIEW

REVIEWER	Adam J Okulicz Kozaryn Rutgers
REVIEW RETURNED	14-Jun-2019

GENERAL COMMENTS	i'm happy--this is good enough for publication would be nice to have computer code to replicate analyses
---

REVIEWER	Alicia Colvin University of Pittsburgh, US
REVIEW RETURNED	30-May-2019

GENERAL COMMENTS	The authors have responded to all of my comments/concerns. I think the paper is improved by the revisions they have made. For the additional analyses they performed including marital status and family history of depression, it might be useful to add to the manuscript text that these additional analyses were performed - but that the results were not significantly altered.
--

REVIEWER	Carla Bezold Detroit Health Department
REVIEW RETURNED	09-Jun-2019

GENERAL COMMENTS	Thank you for making the requested changes
--